# Targeting light-gated chloride channels to neuronal somatodendritic domain reduces their excitatory effect in the axon

**Jessica E Messier[1,2], Hongmei Chen[1,2], Zhao-Lin Cai[1,2], Mingshan Xue[1,2,3]\***

[1]Department of Neuroscience, Baylor College of Medicine, Houston, Texas, United States; [2]The Cain Foundation Laboratories, Jan and Dan Duncan Neurological Research Institute at Texas Children's Hospital, Houston, Texas, United States; [3]Department of Molecular and Human Genetics, Baylor College of Medicine, Houston, Texas, United States

**Abstract** Light-gated chloride channels are emerging as promising optogenetic tools for inhibition of neural activity. However, their effects depend on the transmembrane chloride electrochemical gradient and may be complex due to the heterogeneity of this gradient in different developmental stages, neuronal types, and subcellular compartments. Here we characterized a light-gated chloride channel, GtACR2, in mouse cortical neurons. We found that GtACR2 activation inhibited the soma, but unexpectedly depolarized the presynaptic terminals resulting in neurotransmitter release. Other light-gated chloride channels had similar effects. Reducing the chloride concentrations in the axon and presynaptic terminals diminished the GtACR2-induced neurotransmitter release, indicating an excitatory effect of chloride channels in these compartments. A novel hybrid somatodendritic targeting motif reduced the GtACR2-induced neurotransmitter release while enhancing the somatic photocurrents. Our results highlight the necessity of precisely determining the effects of light-gated chloride channels under specific experimental conditions and provide a much-improved light-gated chloride channel for optogenetic inhibition.

DOI: https://doi.org/10.7554/eLife.38506.001

**\*For correspondence:**
Mingshan.Xue@bcm.edu

**Competing interests:** The authors declare that no competing interests exist.

## Introduction

Targeted manipulation of neural activity is a powerful approach in neuroscience that has provided fundamental insights into the roles of specific neurons in nervous system functions. Genetically encoded actuators such as light-gated ion channels or pumps enable control of neural activity with unprecedented spatiotemporal specificity and are transforming neuroscience research (*Boyden et al., 2005*; *Han and Boyden, 2007*; *Li et al., 2005*; *Nagel et al., 2003*; *Zhang et al., 2007*). Actuators that enable neuronal activation are frequently used, but inhibitory optogenetic tools are increasingly crucial because reversible and temporally precise suppression of neuronal activity is key to revealing the causal roles of specific neurons in network dynamics and behavior. The widely used light-driven inward chloride pumps and outward proton pumps, such as *Natronomonas pharaonis* halorhodopsin (NpHR) and *Halorubrum sodomense* archaerhodopsin (Arch), can hyperpolarize membrane potentials, independent of the electrochemical gradients, to inhibit action potentials with millisecond precision (*Chow et al., 2010*; *Chuong et al., 2014*; *Han and Boyden, 2007*; *Han et al., 2011*; *Zhang et al., 2007*). However, their efficacies are limited because only one ion is transported per absorbed photon, and their activation does not decrease membrane resistance. Light-gated chloride channels, such as *Guillardia theta* anion channelrhodopsin 1 and 2 (GtACR1 and GtACR2), iC++, and iChloC, overcome these limitations (*Berndt et al., 2014*;

**eLife digest** One way to study the role of a specific neuron is to activate or inhibit the cell and then observe the consequences. This can be achieved by using optogenetics, a technique that involves introducing 'light-gated' ion channels in the outer membrane of a target neuron. When light is shone on the cell, these pore-like proteins open their channels: this allows ions to move into or out of the neuron.

Ions flow from high concentration to low concentration areas. Typically, when a neuron is at rest, there are fewer chloride ions inside the cell than outside. Activating a light-gated chloride channel should thus cause these negatively charged ions to enter the neuron. The charge inside of the cell would become more negative relative to the outside: this would inhibit the neuron, making it less likely to fire.

Here, Messier et al. looked into using a light-gated chlorine channel called GtACR2 to inhibit the activity of neurons in mouse brain slices, but the results were not as expected. Activating the chloride channel did inhibit the cell body, the area of the neuron that contains the nucleus. Yet, it had the opposite effect in the axon, the structure that carries electrical signals away from the cell body.

There, activating GtACR2 caused chloride ions to leave the axon, which resulted in the neuron firing. Testing other types of optogenetic chloride channels produced the same result. Further experiments revealed that the concentration of chloride ions is higher inside the axon than the cell body, explaining the observed effects.

Messier et al. then tried to redistribute the channels from the axon to the cell body, where the proteins are inhibitory. This was accomplished by fitting GtACR2 with a molecular tag that acts like an address label, with the cell body as the target destination. Overall, when these modified channels were activated, the neuron was more strongly inhibited.

Ultimately, the GtACR2 channel designed by Messier et al. is a powerful new inhibitory optogenetic tool. In addition, this tool could be used to study chloride gradients in brain regions, cell types and areas of cells that are otherwise difficult to access.

DOI: https://doi.org/10.7554/eLife.38506.002

Govorunova et al., 2015, 2017b; Wietek et al., 2014). They are highly sensitive to light, allow multiple ions to cross the membrane per photocycle, and reduce membrane resistance, thereby potently inhibiting action potentials. Thus, light-gated chloride channels are emerging as promising optogenetic tools for suppressing neuronal activity (Govorunova et al., 2017a).

Determining the precise effects of light-gated ion channels or pumps under defined conditions is a prerequisite to use them for interrogating the functions of specific neurons and circuits. This is due to the possibility that these channels or pumps not only modulate membrane potentials but also may affect other processes such as ion homeostasis or neurotransmitter release. For example, activation of a light-driven inward chloride pump, eNpHR3.0, can transiently change the reversal potential of $GABA_A$ receptors and alter the inhibitory synaptic inputs (Raimondo et al., 2012). Prolonged activation of a light-driven outward proton pump, eArch3.0, can increase presynaptic calcium concentrations and spontaneous neurotransmitter release (Mahn et al., 2016). Despite these potential confounds, these inhibitory optogenetic molecules become increasingly important tools for targeted silencing of neuronal populations, as long as these confounds are understood and controlled (Allen et al., 2015). Therefore, it is also crucial to thoroughly characterize light-gated chloride channels, as their effect depends on the difference between the membrane potential and the reversal potential for chloride, both of which can vary in different neuronal types and subcellular compartments (Marty and Llano, 2005; Trigo et al., 2008).

To this end, we investigated the effects of activating GtACR2 in mouse cortical excitatory and inhibitory neurons. Much to our surprise, wide-field light activation of GtACR2 not only inhibited the soma, but also caused neurotransmitter release onto neighboring neurons. A similar phenomenon was observed with GtACR1, iC++, and iChloC. We further showed that GtACR2 activation in the axon and presynaptic terminals directly depolarized the membrane to induce neurotransmitter release due to high chloride concentrations in these compartments. These data explain the recent

observations that photostimulation of neurons expressing GtACR1 or GtACR2 can paradoxically release neurotransmitters or generate action potentials (*Mahn et al., 2016*; *Malyshev et al., 2017*). To reduce the excitatory effect of GtACR2, we screened a panel of somatodendritic targeting motifs to reduce the trafficking of GtACR2 to the axon and presynaptic terminals. We created a hybrid motif (Kv2.1C-linker-TlcnC) that is most effective in concentrating GtACR2 in the somatodendritic domain. Activation of somatodendritically targeted GtACR2 resulted in larger photocurrents at the soma and less neurotransmitter release than wild type GtACR2. Thus, restricting the localization of light-gated chloride channels to the somatodendritic domain improves the inhibitory efficacy of these optogenetic tools. Our results also suggest that light-gated chloride channels can be a useful tool for studying the physiological functions of chloride electrochemical gradients in specific brain regions, cell types, or subcellular domains that are otherwise difficult to access, such as small axons and presynaptic terminals.

## Results

### Light activation of GtACR2 in mouse cortical neurons causes neurotransmitter release

To examine the efficacy of GtACR2 in mouse cortical excitatory neurons, we expressed a GtACR2-EYFP fusion protein (referred to as GtACR2 below) together with a red fluorescent protein, tdTomato, in layer 2/3 pyramidal neurons of the mouse visual or somatosensory cortex by *in utero* electroporation of plasmids at embryonic day 14.5–15.5. We obtained acute coronal brain slices from 3 to 8 week-old mice and observed that GtACR2 was present in the soma, dendrites, and axon (*Figure 1A*). We performed whole-cell patch clamp recordings at the soma of neurons expressing GtACR2 (GtACR2$^+$ neurons) with a K$^+$-based pipette solution (*Figure 1B*). As previously reported (*Govorunova et al., 2015*), activation of GtACR2 by wide-field blue light (455 nm) potently inhibited current-induced spiking in these neurons (*Figure 1C*). However, when we voltage clamped the neurons to record GtACR2-mediated photocurrents, we unexpectedly found an inward current that was superimposed on the photocurrent. This inward current resembled an excitatory postsynaptic current (EPSC; *Figure 1D*). To further investigate this phenomenon, we recorded layer 2/3 pyramidal neurons that did not express GtACR2 (GtACR2$^-$ neurons) with a Cs$^+$-based pipette solution (*Figure 1E*). Activation of GtACR2 with a short light pulse (0.5–10 ms) generated inward currents in all recorded GtACR2$^-$ neurons that were voltage clamped at the reversal potential for GABAergic inhibition (−60 mV). The onsets of these inward currents followed the onset of the blue light by 3.19 ± 0.26 ms (mean ± s.e.m., $n$ = 25). These currents were abolished by the glutamatergic receptor antagonists, NBQX and CPP (*Figure 1F*), or the voltage-gated sodium channel blocker, tetrodotoxin (TTX; *Figure 1G*), indicating that they were indeed monosynaptic EPSCs caused by the glutamate transmitter released from GtACR2$^+$ neurons. Activation of GtACR2 also produced inhibitory postsynaptic currents (IPSCs) in GtACR2$^-$ neurons that were voltage clamped at the reversal potential for glutamatergic excitation (+10 mV). These IPSCs were disynaptic because they were abolished by NBQX and CPP (*Figure 1—figure supplement 1*), indicating that activating GtACR2 in pyramidal neurons can release sufficient glutamate to recruit inhibitory interneurons.

To determine if the phenomenon of GtACR2-induced neurotransmitter release also occurs in GABAergic inhibitory neurons, we expressed GtACR2 in parvalbumin-expressing (Pv) neurons by injecting a Flpo recombinase-dependent adeno-associated virus (AAV) into the visual cortex of *Pvalb-2A-Flpo* mice (*Pvalb$^{Flpo/+}$*) (*Madisen et al., 2015*) at postnatal day 1 (*Figure 1H*). Using acute brain slices from 3 to 6 week-old mice, we found that activation of GtACR2 in Pv neurons generated IPSCs in all recorded GtACR2$^-$ layer 2/3 pyramidal neurons, and the IPSCs were abolished by Gabazine, a GABA$_A$ receptor antagonist, or TTX (*Figure 1I–K*). The onsets of the IPSCs followed the onset of the blue light by 2.47 ± 0.15 ms (mean ± s.e.m., $n$ = 17), indicating that they were monosynaptic IPSCs caused by the GABA transmitter released from GtACR2$^+$ Pv neurons.

We next determined how repetitive or prolonged activation of GtACR2 would affect neurotransmitter release. We activated GtACR2 in layer 2/3 pyramidal neurons or Pv neurons with a high-frequency train of light pulses and found that each light pulse produced reliable EPSCs or IPSCs, respectively (*Figure 1—figure supplement 2*). Interestingly, continuous activation of GtACR2 in layer 2/3 pyramidal neurons or Pv neurons with a long pulse of light (e.g., 2 s) transiently generated

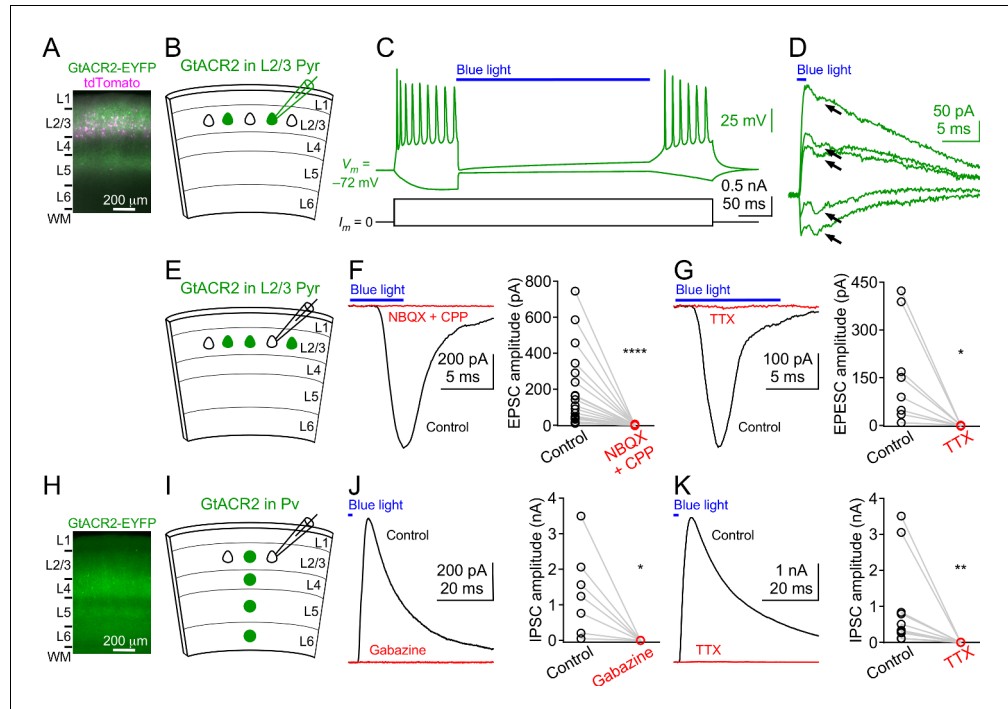

**Figure 1.** Light activation of GtACR2 causes excitatory and inhibitory neurons to release neurotransmitters. (**A**) A representative fluorescent image of the visual cortex showing GtACR2-EYFP and tdTomato expression in a subset of layer 2/3 pyramidal neurons. Note the strong EYFP fluorescence in layer 5 that contains the axons of layer 2/3 pyramidal neurons. L, layer; WM, white matter. (**B**) Schematic of slice experiments in (**C,D**). GtACR2 in a subset of layer 2/3 pyramidal neurons. (**C**) Membrane potentials (upper panel) in response to somatic current injections (lower panel) from a GtACR2$^+$ neuron. Blue light activation of GtACR2 suppressed the action potentials evoked by current injections ($n = 5$ neurons). (**D**) Blue light-induced membrane currents recorded at the membrane potentials of $-75$, $-77$, $-78$, $-79$, and $-80$ mV from the same GtACR2$^+$ neuron in (**C**). Note the EPSC-like inward currents (arrows) superimposed on the GtACR2-mediated photocurrents ($n = 2$ neurons). (**E**) Schematic of slice experiments in (**F,G**). GtACR2 in a subset of layer 2/3 pyramidal neurons. (**F**) Left, photoactivation of GtACR2 generated EPSCs in a GtACR2$^-$ neuron, which were abolished by the glutamate receptor antagonists, NBQX and CPP. Right, summary graph of similar experiments ($n = 17$, p<0.0001). (**G**) Left, photoactivation of GtACR2 generated EPSCs in a GtACR2$^-$ neuron, which were abolished by the voltage-gated sodium channel blocker, TTX. Right, summary graph of similar experiments ($n = 8$, p=0.02). (**H**) A representative fluorescent image of the visual cortex showing GtACR2-EYFP expression in Pv neurons. (**I,J,K**) As in (**E,F,G**), but for GtACR2 in Pv neurons. GtACR2 activation-induced IPSCs were abolished by the GABA$_A$ receptor antagonist, Gabazine (**J**, $n = 7$, p=0.02) or TTX (**K**, $n = 10$, p=0.002).

DOI: https://doi.org/10.7554/eLife.38506.003

The following figure supplements are available for figure 1:

**Figure supplement 1.** Recruitment of inhibitory neurons by activating GtACR2 in excitatory neurons.

DOI: https://doi.org/10.7554/eLife.38506.004

**Figure supplement 2.** Neurotransmitter release induced by repetitive activation of GtACR2.

DOI: https://doi.org/10.7554/eLife.38506.005

**Figure supplement 3.** Neurotransmitter release induced by prolonged activation of GtACR2.

DOI: https://doi.org/10.7554/eLife.38506.006

large EPSCs or IPSCs, respectively, followed by various amounts of smaller synaptic responses (*Figure 1—figure supplement 3A,B,E,F,G*). Typically, more than 50% of the neurotransmitter release occurred within the first 100–200 ms of the light pulse (*Figure 1—figure supplement 3C,D,H,I*).

The finding of GtACR2-induced neurotransmitter release was unexpected, because the Nernst equilibrium potential of chloride becomes lower than the action potential threshold in rodent cortical neurons after the second postnatal week, and activation of chloride channels should inhibit, not promote, neurotransmitter release onto neighboring cells (*Ben-Ari, 2002*; *Owens et al., 1996*). We

thus sought to identify the cause of this paradoxical neurotransmitter release and considered three possibilities. First, GtACR2 may conduct cations to depolarize neurons. Second, an increase in the intracellular chloride or strong hyperpolarization induced by GtACR2 activation may lead to rebound spikes. Third, GtACR2-mediated chloride currents may be excitatory.

## A GtACR2-mediated excitatory chloride conductance causes neurotransmitter release

First, it was reported that GtACR2 did not conduct physiological cations (*Govorunova et al., 2015*), but some other light-gated chloride channels retained certain cation conductance (*Berndt et al., 2014*; *Wietek et al., 2014*). Thus, we sought to verify that in cortical neurons, GtACR2 has a similar reversal potential as a known chloride channel. To accomplish this, we determined the reversal potential of GtACR2-mediated photocurrents in comparison with that of IPSCs mediated by the endogenous $GABA_A$ receptors. We used *in utero* electroporation to express GtACR2 in layer 2/3 pyramidal neurons and a Cre recombinase-dependent AAV to express a red light-gated cation channelrhodopsin, ReaChR, in Pv neurons of *Pvalb-2A-Cre* mice ($Pvalb^{Cre/+}$) (*Madisen et al., 2010*). We performed whole-cell voltage clamp recordings at the soma of a $GtACR2^+$ layer 2/3 pyramidal neuron and a nearby $GtACR2^-$ pyramidal neuron simultaneously (*Figure 2—figure supplement 1A*). In the $GtACR2^+$ neuron, we sequentially recorded the IPSCs induced by activating ReaChR in Pv neurons via 617 nm light and the GtACR2-mediated photocurrents activated by 455 nm light. Both IPSCs and photocurrents were recorded at different membrane potentials to determine their reversal potentials in the same neuron (*Figure 2—figure supplement 1B,C*). 617 nm light does not activate GtACR2 (*Govorunova et al., 2015*), whereas 455 nm light partially activates ReaChR (*Lin et al., 2013*). Thus, to avoid Pv neuron-mediated IPSCs contaminating GtACR2-mediated photocurrents, we monitored the IPSCs in the $GtACR2^-$ neuron at the membrane potential of +10 mV to ensure that the intensity of the 455 nm light was not sufficient to activate Pv neurons and generate IPSCs (*Figure 2—figure supplement 1B*). We found that the reversal potentials of GtACR2-mediated photocurrents and GABAergic IPSCs were similar for each neuron and well below the action potential threshold (*Figure 2—figure supplement 1D*), indicating that GtACR2 does not conduct cations to cause neurotransmitter release.

Second, we tested if GtACR2-induced neurotransmitter release could be due to rebound depolarization. An increase in the intracellular chloride caused by GtACR2 activation may trigger rapid efflux of chloride after the blue light illumination terminates. However, this possibility is unlikely because when GtACR2 was activated by a long pulse of blue light, neurotransmitter release occurred before the light illumination ended (see examples in *Figure 1F,G* and *Figure 1—figure supplement 3*). Another possibility is that the strong hyperpolarization induced by GtACR2 activates hyperpolarization-activated $I_h$ currents, which may depolarize the membrane potential above the action potential threshold. However, pharmacological inhibition of $I_h$ currents slightly increased the amplitudes of GtACR2-induced EPSCs (*Figure 2—figure supplement 2A,B*), most likely because inhibiting $I_h$ currents increases neuronal membrane resistances (*Robinson and Siegelbaum, 2003*). Thus, GtACR2-induced neurotransmitter release is not caused by rebound depolarization.

Third, although GtACR2-mediated photocurrents are inhibitory at the soma, it is possible that the chloride concentrations are higher in some other cellular compartments, such that the electrochemical gradient causes chloride to exit the cell upon GtACR2 channel opening, resulting in depolarization of the membrane potential. To test this hypothesis, we pharmacologically inhibited the activity of $Na^+$-$K^+$-$2Cl^-$ cotransporter 1 (NKCC1) with bumetanide (50 or 100 μM) to decease the intracellular chloride concentrations, as NKCC1 is responsible for transporting chloride into neurons (*Ben-Ari, 2017*). When we activated GtACR2 in layer 2/3 pyramidal neurons, the resulting EPSCs in $GtACR2^-$ pyramidal neurons were diminished by bath application of bumetanide (*Figure 2A,B*), indicating that GtACR2-induced neurotransmitter release requires high concentrations of intracellular chloride. An alternative interpretation of this result would be that bumetanide blocks GtACR2 itself. To test this possibility, we simultaneously recorded the photocurrents and EPSCs in $GtACR2^+$ and $GtACR2^-$ pyramidal neurons, respectively. While bumetanide diminished the EPSCs in $GtACR2^-$ neurons, it had no effect on the photocurrents in $GtACR2^+$ neurons (*Figure 2—figure supplement 3A, B*), thereby ruling out the possibility that bumetanide affects GtACR2 itself. Furthermore, when a cation channel, channelrhodopsin-2 (ChR2) (*Boyden et al., 2005*; *Li et al., 2005*; *Nagel et al., 2003*), was expressed and activated in layer 2/3 pyramidal neurons, the resulting EPSCs in $ChR2^-$

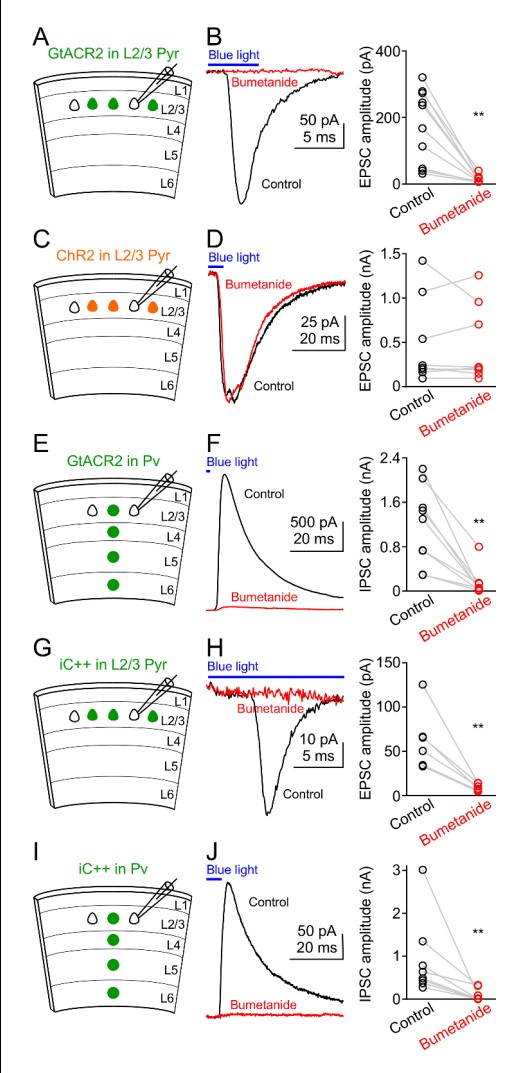

**Figure 2.** Reducing intracellular chloride concentrations diminishes the neurotransmitter release induced by activation of GtACR2 or iC++. (**A**) Schematic of slice experiments in (**B**). GtACR2 in a subset of layer 2/3 pyramidal neurons. (**B**) Left, photoactivation of GtACR2 generated EPSCs in a GtACR2⁻ neuron, which were diminished by decreasing intracellular chloride concentrations with an NKCC1 blocker, bumetanide. Right, summary graph of similar experiments ($n$ = 10, p=0.002). (**C,D**) As in (**A,B**), but for ChR2. ChR2 activation-induced EPSCs were not affected by bumetanide ($n$ = 8, p=0.9). (**E,F**) As in (**A,B**), but for GtACR2 in Pv neurons. GtACR2 activation-induced IPSCs were diminished by bumetanide ($n$ = 9, p=0.004). (**G,H**) As in (**A,B**), but for iC++. iC++-induced EPSCs were diminished by bumetanide ($n$ = 6, p=0.009). (**I,J**) As in (**E,F**), but for iC++. iC++-induced IPSCs were diminished by bumetanide ($n$ = 8, p=0.008).
DOI: https://doi.org/10.7554/eLife.38506.007

The following figure supplements are available for figure 2:

*Figure 2 continued on next page*

neurons were not affected by bumetanide (*Figure 2C,D*), indicating that reducing the intracellular chloride concentration only affects chloride-mediated, and not cation-mediated, EPSCs. Finally, bumetanide also diminished the IPSCs resulting from GtACR2 activation in Pv neurons (*Figure 2E,F*). Together, these results demonstrate that light activation of GtACR2 generates an excitatory chloride conductance in certain neuronal compartments to trigger neurotransmitter release.

## Expression and activation of GtACR2 in adult neurons cause neurotransmitter release

Since we expressed GtACR2 by *in utero* electroporation or neonatal AAV injection, we sought to determine if the excitatory effect of GtACR2 was caused by the long-term expression of GtACR2 throughout development that somehow altered the chloride homeostasis. To selectively express GtACR2 in adult neurons, we *in utero* electroporated a Flpo-dependent plasmid into layer 2/3 pyramidal neurons that will express GtACR2 only if Flpo is present (*Figure 2—figure supplement 2C,D*). A Flpo-expressing AAV was then injected into the electroporated mice at postnatal week 4 or 9 to turn on the GtACR2 expression. We obtained acute coronal brain slices 1–3 weeks after injecting Flpo-expressing AAV and found that light activation of GtACR2, again, produced bumetanide-sensitive EPSCs in GtACR2⁻ neurons (*Figure 2—figure supplement 2E*). Thus, it is unlikely that GtACR2 expression during neuronal development alters the chloride homeostasis to render GtACR2 excitatory, as acute expression of GtACR2 in mature neurons had the same effect.

## Activation of other light-gated chloride channels causes neurotransmitter release

To determine if activation of other light-gated chloride channels can trigger neurotransmitter release, we examined iC++ and iChloC, two engineered blue light-gated chloride channels that were converted from cation channelrhodopsins (*Berndt et al., 2016*; *Wietek et al., 2015*), and GtACR1, another natural anion channelrhodopsin from *Guillardia theta* (*Govorunova et al., 2015*). We *in utero* electroporated plasmids to express iC++ in layer 2/3 pyramidal neurons of the visual cortex and obtained acute coronal brain slices from 3 to 9 week-old mice. Similar to GtACR2, light activation of iC++ generated EPSCs in iC++⁻ layer 2/3 pyramidal neurons that

*Figure 2 continued*

**Figure supplement 1.** The reversal potential of GtACR2 at the soma is similar to that of GABA_A receptors.
DOI: https://doi.org/10.7554/eLife.38506.008

**Figure supplement 2.** GtACR2-induced neurotransmitter release is not due to rebound depolarization or its long-term expression throughout development.
DOI: https://doi.org/10.7554/eLife.38506.009

**Figure supplement 3.** Bumetanide does not affect GtACR2 and iC++-mediated photocurrents.
DOI: https://doi.org/10.7554/eLife.38506.010

were abolished by NBQX and CPP (amplitude reduced by $97.3 \pm 0.9\%$, mean $\pm$ s.e.m., $n = 4$). Bumetanide diminished the iC++-induced EPSCs (*Figure 2G,H*) without affecting the iC++-mediated photocurrents (*Figure 2—figure supplement 3C,D*). We also expressed iC++ in Pv neurons by injecting a Cre-dependent AAV into $Pvalb^{Cre/+}$ mice and found that light activation of iC++ caused bumetanide-sensitive IPSCs in iC++$^-$ layer 2/3 pyramidal neurons (*Figure 2I,J*). Similarly, when we expressed iChloC in Pv neurons, light activation of iChloC resulted in IPSCs in 10 out of 17 recorded iChloC$^-$ layer 2/3 pyramidal neurons ($183 \pm 42$ pA, mean $\pm$ s.e.m., $n = 10$), presumably because iChloC generated smaller photocurrents than iC++ (iChloC, $292 \pm 62$ pA, $n = 7$; iC++, $2182 \pm 291$ pA, $n = 15$; recorded at the membrane potential of $+10$ mV; mean $\pm$ s.e.m., p<0.0001, $t$ test with Welch's correction). Finally, activation of GtACR1 in layer 2/3 pyramidal neurons produced EPSCs onto all recorded GtACR1$^-$ pyramidal neurons ($149 \pm 47$ pA, mean $\pm$ s.e.m., $n = 11$). Altogether, these results demonstrate that activation of different light-gated chloride channels in neurons can trigger neurotransmitter release.

## GtACR2 activation directly depolarizes the presynaptic terminals

We hypothesized that the most likely neuronal compartments rendering GtACR2 excitatory were the distal axon and presynaptic terminals because of the following previous findings. First, activation of presynaptic GABA_A or glycine receptors enhanced neurotransmitter release at several synapses of the hippocampus, cerebellum, and brainstem (*Jang et al., 2006*; *Pugh and Jahr, 2011*; *Ruiz et al., 2010*; *Stell et al., 2007*; *Turecek and Trussell, 2001*; *Zorrilla de San Martin et al., 2017*). Second, the chloride concentrations were 4–5 times higher in the presynaptic terminals of the Calyx of Held than the parent soma (*Price and Trussell, 2006*). Third, there appeared to be an axo-somato-dendritic gradient in which the reversal potentials of GABA from the axon to the soma and dendrites of cortical neurons become progressively more negative (*Khirug et al., 2008*). To test our hypothesis, we expressed GtACR2 in layer 2/3 pyramidal neurons of the visual cortex in one hemisphere as described above and obtained acute coronal slices from the contralateral hemisphere (*Figure 3A*). GtACR2 was present in the long-range callosal projections in the contralateral hemisphere (*Figure 3—figure supplement 1A*), which enabled us to activate GtACR2 in the axon and presynaptic terminals that were severed from their parent somas. Light activation of GtACR2 in the callosal projections generated EPSCs in layer 2/3 pyramidal neurons of the contralateral cortex, which were diminished by TTX (*Figure 3—figure supplement 1B,C*) or bumetanide (*Figure 3B*). These results demonstrate that activation of GtACR2 in the axon and presynaptic terminals is sufficient to trigger neurotransmitter release.

If GtACR2-mediated chloride currents are excitatory in the presynaptic terminals, then GtACR2 should be similar to ChR2, whose activation can directly depolarize the presynaptic membrane in the absence of action potentials to trigger neurotransmitter release (*Petreanu et al., 2009*). To test this prediction, we recorded EPSCs or IPSCs in GtACR2$^-$ neurons while activating GtACR2 in layer 2/3 pyramidal neurons or Pv neurons, respectively (*Figure 3C,E*). As described above, bath application of TTX abolished the EPSCs and IPSCs. However, when we further blocked voltage-gated potassium channels by 4-aminopyridine (4-AP) and tetraethylammonium (TEA) to prolong membrane depolarization (*Petreanu et al., 2009*), the EPSCs and IPSCs were partially recovered (*Figure 3D,F*). These results indicate that in the absence of action potentials, light activation of GtACR2 is sufficient to depolarize the presynaptic membrane to open voltage-gated calcium channels and trigger neurotransmitter release.

We further tested if GtACR2-induced axonal depolarization could evoke antidromic action potentials by performing extracellular loose-patch or whole-cell current clamp recordings at the somas of GtACR2$^+$ pyramidal neurons. We observed antidromic spikes in 9 out of 88 neurons recorded in loose-patch configuration and 21 out of 56 neurons recorded in whole-cell configuration in response

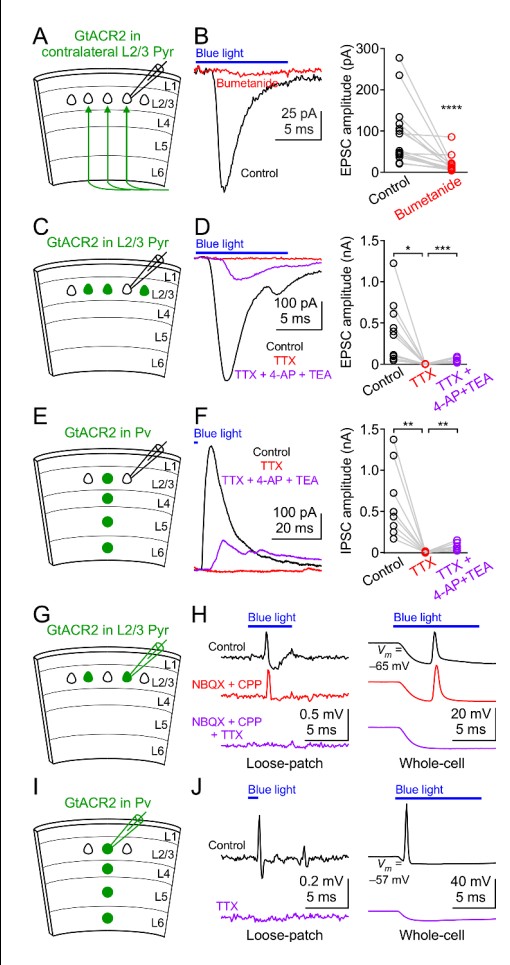

**Figure 3.** GtACR2 activation directly depolarizes the distal axon and presynaptic terminals and can result in antidromic spikes. (**A**) Schematic of slice experiments in (**B**). GtACR2 in a subset of layer 2/3 pyramidal neurons in the contralateral hemisphere. (**B**) Left, photoactivation of GtACR2 in the callosal axons, severed from their somas, generated EPSCs in a GtACR2⁻ neuron, which were diminished by bumetanide. Right, summary graph of similar experiments (*n* = 15, p<0.0001). (**C**) Schematic of slice experiments in (**D**). GtACR2 in a subset of layer 2/3 pyramidal neurons. (**D**) Left, photoactivation of GtACR2 generated EPSCs in a GtACR2⁻ neuron, which were abolished by TTX, but partially recovered by the addition of voltage-gated potassium channel blockers, 4-AP (1.5 mM) and TEA (1.5 mM). Right, summary graph of similar experiments (*n* = 10; TTX vs. control, p=0.02, average EPSC amplitude in TTX was 2% of control; TTX + 4-AP + TEA vs. TTX, p=0.0004, average EPSC amplitude in TTX + 4-AP + TEA was 35% of control). (**E,F**) As in (**C,D**), but for Pv neurons. GtACR2 activation-induced IPSCs were abolished by TTX, but partially recovered by 4-AP and TEA (*n* = 9; TTX vs. control, p=0.006, average IPSC amplitude in TTX was 0.9% of control; TTX + 4-AP + TEA vs. TTX, p=0.008,
*Figure 3 continued on next page*

to blue light stimulation (*Figure 3G,H*). In the whole-cell current clamp recordings, although the chloride concentration in the patch pipette solution sets the Nernst equilibrium potential of chloride around −85 mV (see Materials and methods), blue light induced a depolarization following the initial hyperpolarization. This observation is consistent with the notion that the depolarization antidromically propagated from the distal axon to the soma (*Figure 3H*). The antidromic spikes were not affected by NBQX and CPP, but were abolished by TTX (*Figure 3H*), indicating that the spikes were generated within the GtACR2⁺ neurons, rather than by excitatory inputs from other neurons. Antidromic spikes were only observed in a subset of neurons, likely because the GtACR2 expression levels are heterogeneous in different neurons, and the hyperpolarization initiated at the soma can orthodromically propagate to counteract the antidromic spikes. Similarly, TTX-sensitive antidromic spikes were observed in a subset of GtACR2⁺ Pv neurons (3 out of 11 neurons in loose-patch configuration and 7 out of 17 neurons in whole-cell configuration, *Figure 3I,J*). These results show that GtACR2-induced axonal depolarization can be sufficient to elicit antidromic action potentials.

## Targeting GtACR2 to the somatodendritic domain reduces light-induced neurotransmitter release

Activating GtACR2 and other light-gated chloride channels inhibits the soma but depolarizes the presynaptic terminals to release neurotransmitters. This dichotomic effect can confound the utilization of these channels as inhibitory optogenetic tools for suppressing neuronal activity. We reasoned that reducing the trafficking of light-gated chloride channels into the axon and presynaptic terminals should reduce or eliminate their depolarizing action. Thus, we sought to restrict GtACR2 within the somatodendritic domain of neurons by fusing GtACR2 with a number of reported somatodendritic targeting motifs including a 26-amino acid Myosin Va-binding domain of Melanophilin (MBD) (*Lewis et al., 2009*), a 32-amino acid cytoplasmic C-terminal motif of Neuroligin 1 (Nlgn1C) (*Rosales et al., 2005*), a 16-amino acid dileucine-containing motif of potassium channel Kv4.2 (Kv4.2LL) (*Rivera et al., 2003*), the N-terminal 150 residues of kainate receptor subunit 2 (KA2N) (*Shemesh et al., 2017*), the C-terminal 17 residues of Telencephalin (TlcnC) (*Mitsui et al.,*

*Figure 3 continued*

average IPSC amplitude in TTX + 4-AP + TEA was 23% of control). (**G**) Schematic of slice experiments in (**H**). GtACR2 in a subset of layer 2/3 pyramidal neurons. (**H**) Photoactivation of GtACR2 generated antidromic spikes in GtACR2+ pyramidal neurons, which were not affected by NBQX and CPP, but were abolished by TTX in both loose-patch (left panel) and whole-cell (right panel) recordings. In the whole-cell recordings, the resting membrane potentials of those neurons that generated antidromic spikes were −68.1 ± 1.7 mV (mean ± s.e.m., *n* = 21). (**I,J**) As in (**G,H**), but for Pv neurons. The antidromic spikes in GtACR2+ Pv neurons were abolished by TTX. In the whole-cell recordings, the resting membrane potentials of those neurons that generated antidromic spikes were −60.0 ± 2.4 mV (mean ± s.e.m., *n* = 7).

DOI: https://doi.org/10.7554/eLife.38506.011

The following figure supplement is available for figure 3:

**Figure supplement 1.** Activation of GtACR2 in the callosal axons causes neurotransmitter release.
DOI: https://doi.org/10.7554/eLife.38506.012

2005), and a 65-amino acid cytoplasmic C-terminal motif of potassium channel Kv2.1 (Kv2.1C) (*Lim et al., 2000*; *Wu et al., 2013*). Each of these GtACR2 variants (*Table 1*), along with tdTomato, were expressed in layer 2/3 pyramidal neurons of the visual cortex by *in utero* electroporation (*Figure 4A*). Since GtACR2 was tagged with EYFP or EGFP, we compared the EYFP or EGFP fluorescence in layer 5, which only contains the axons of layer 2/3 pyramidal neurons, with the EYFP or EGFP fluorescence in layer 2/3 to estimate the distribution of GtACR2 between the axon and somatodendritic domain. We normalized the EYFP or EGFP fluorescence ratio between layer 5 and layer 2/3 by the tdTomato fluorescence ratio between layer 5 and layer 2/3 to control for variations in the collateral axons. Among tested motifs, TlcnC and Kv2.1C were most effective in targeting GtACR2 to the soma and dendrites (*Figure 4D*). As these two motifs may engage different trafficking mechanisms, we combined them to create two hybrid motifs, Kv2.1C-TlcnC and Kv2.1C-linker-TlcnC. Kv2.1C-linker-TlcnC turned out to be the best in restricting GtACR2 within the somatodendritic domain

(*Figure 4A,D*). Interestingly, GtACR2-EYFP-Kv2.1C and GtACR2-EYFP-Kv2.1C-linker-TlcnC showed less intracellular aggregation than wild type GtACR2 (*Figure 4B*), suggesting that the somatodendritic targeting motifs also enhance the surface expression of GtACR2. Finally, the EYFP fluorescence of the callosal projections in the contralateral hemisphere was also reduced for GtACR2-EYFP-Kv2.1C and GtACR2-EYFP-Kv2.1C-linker-TlcnC as compared to GtACR2-EYFP (*Figure 4C,E*), demonstrating that both somatodendritic targeting motifs decreased the trafficking of GtACR2 into the distal axon.

To determine how targeting GtACR2 to the somatodendritic domain affects its photocurrent and ability to trigger neurotransmitter release, we compared the somatodendritically targeted GtACR2

**Table 1.** Amino acid sequences of somatodendritic targeting motifs.
Black bold, amino acid sequences of somatodendritic targeting motifs; blue, GtACR2; red, EYFP or EGFP; purple, linker.

| Motif | Sequence |
| --- | --- |
| MBD | *GtACR2*-*EYFP*-GSGSGTRGSGS-**RDQPLNSKKKKRLLSFRDVDFEEDSD** |
| Nlgn1C | *GtACR2*-*EYFP*-**VVLRTACPPDYTLAMRRSPDDIPLMTPNTITM** |
| Kv4.2LL | *GtACR2*-*EYFP*-**FETQHHHLLHCLEKTT** |
| KA2N | *GtACR2*-GGSGGTGGSGGT-**MPAELLLLLIVAFANPSCQVLSSLRMAAILDDQTVCGRGERLALALAREQINGIIEVPAKARVEVDIFELQRDSQYETTDTMCQILPKGVVSVLGPSSSPASASTVSHICGEKEIPHIKVGPEETPRLQYLRFASVSLYPSNEDVSLAVS**-GASGGT-*EGFP* |
| TlcnC | *GtACR2*-*EYFP*-**AESPADGEVFAIQLTSS** |
| Kv2.1C | *GtACR2*-*EYFP*-**QSQPILNTKEMAPQSKPPEELEMSSMPSPVAPLPARTEGVIDMRSMSSIDSFISCATDFPEATRF** |
| Kv2.1C-TlcnC | *GtACR2*-*EYFP*-**QSQPILNTKEMAPQSKPPEELEMSSMPSPVAPLPARTEGVIDMRSMSSIDSFISCATDFPEATRF**-**AESPADGEVFAIQLTSS** |
| Kv2.1C-linker-TlcnC | *GtACR2*-*EYFP*-**QSQPILNTKEMAPQSKPPEELEMSSMPSPVAPLPARTEGVIDMRSMSSIDSFISCATDFPEATRF**-GSGSGSGSGS-**AESPADGEVFAIQLTSS** |

DOI: https://doi.org/10.7554/eLife.38506.014

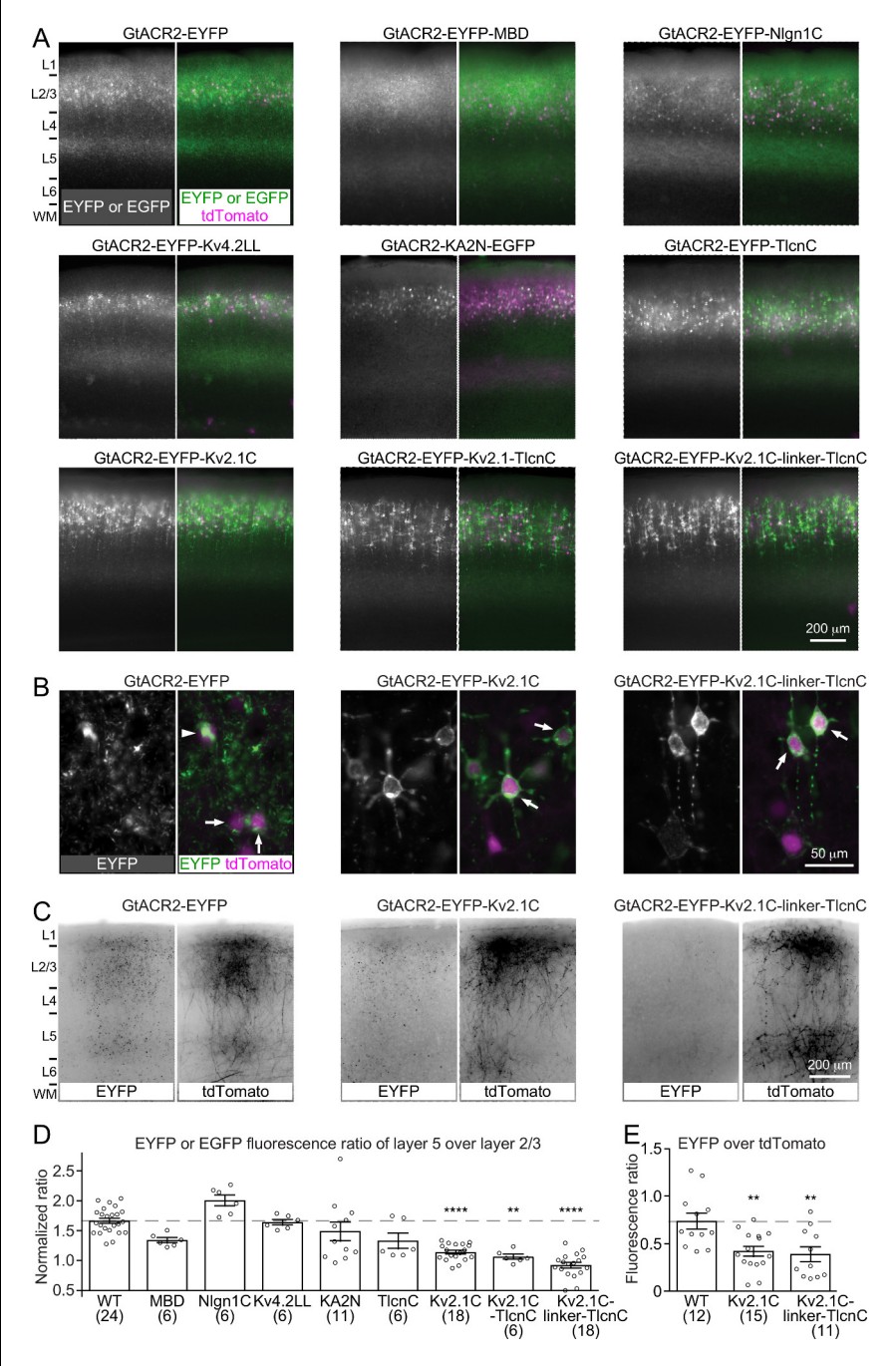

**Figure 4.** Targeting GtACR2 to neuronal somatodendritic domain. (**A**) Wild type (WT) GtACR2 and its variants tagged with EYFP or EGFP were expressed along with tdTomato in a subset of layer 2/3 pyramidal neurons. Representative fluorescent images of the cortices showing the distribution of GtACR2. Note the strong EYFP fluorescence in layer 5 for WT GtACR2 and weak EYFP fluorescence in layer 5 for some somatodendritically targeted GtACR2 variants (e.g., Kv2.1C-linker-TlcnC). L, layer; WM, white matter. (**B**) Representative fluorescent images of electroporated neurons showing that GtACR2-EYFP-Kv2.1C and GtACR2-EYFP-Kv2.1C-linker-TlcnC were more concentrated on the membranes at the soma and proximal dendrites (arrows) than GtACR2-EYFP. Note the intracellular aggregation of GtACR2-EYFP (arrow head). (**C**) Representative fluorescent images of the contralateral hemisphere showing less GtACR2-EYFP-Kv2.1C and GtACR2-EYFP-Kv2.1C-linker-TlcnC in the callosal projections than GtACR2-EYFP. The callosal projections were labeled by tdTomato. (**D**) Summary data of the experiments in (**A**). The EYFP or EGFP fluorescence ratio between layer 5 and layer 2/3 was normalized by the tdTomato fluorescence ratio between layer 5 and layer 2/3. A reduction in the normalized EYFP or EGFP fluorescence ratio

*Figure 4 continued on next page*

*Figure 4 continued*

indicates a shift of the EYFP or EGFP distribution from the axon to somatodendritic domain. (**E**) Summary data of the experiments in (**C**). The ratio of EYFP fluorescence to tdTomato fluorescence of the callosal projections in the contralateral hemisphere. A reduction in the ratio indicates less EYFP in the callosal axons. The numbers of analyzed slices were indicated in the panels. The columns and error bars are mean ± s.e.m. **p<0.01, ****p<0.0001.

DOI: https://doi.org/10.7554/eLife.38506.013

variants, GtACR2-EYFP-Kv2.1C and GtACR2-EYFP-Kv2.1C-linker-TlcnC, with wild type GtACR2 in the same litters of mice. We first recorded GtACR2$^+$ layer 2/3 pyramidal neurons and found that the blue light-activated photocurrents of GtACR2-EYFP-Kv2.1C and GtACR2-EYFP-Kv2.1C-linker-TlcnC were 2.1–2.4 and 2.7–3.5 folds of GtACR2-EYFP photocurrents, respectively (*Figure 5A–C* and *Figure 5—figure supplement 1A,C,E,G*). We then recorded the EPSCs in GtACR2$^-$ layer 2/3 pyramidal neurons in response to different strengths of blue light stimulation. The EPSCs evoked by activating GtACR2-EYFP-Kv2.1C or GtACR2-EYFP-Kv2.1C-linker-TlcnC were reduced by 52–60% or 65–77%, respectively, as compared to those evoked by activating GtACR2-EYFP (*Figure 5D–G* and *Figure 5—figure supplement 1B,D,F,H*). In a subset of experiments with GtACR2-EYFP-Kv2.1C or GtACR2-EYFP-Kv2.1C-linker-TlcnC, the lowest light stimulation strength could no longer evoke EPSCs in GtACR2$^-$ neurons, but still induced robust photocurrents in GtACR2$^+$ neurons (*Figure 5—figure supplement 1E–H*). Furthermore, when photostimulating the callosal projections, the EPSCs in layer 2/3 pyramidal neurons of the contralateral cortex were reduced by 54% and 73% for GtACR2-EYFP-Kv2.1C and GtACR2-EYFP-Kv2.1C-linker-TlcnC, respectively, as compared to GtACR2-EYFP (*Figure 5H,I*). These results show that the somatodendritic targeting motifs, especially Kv2.1C-linker-TlcnC, shift GtACR2 from the axon towards the soma and dendrites, thereby reducing the excitatory action in the axon and presynaptic terminals while enhancing the inhibitory currents at the soma and dendrites.

## Discussion

Optogenetic suppression of neuronal activity and synaptic outputs is an essential approach for dissecting the roles of specific neurons in brain functions. Light-gated chloride channels, particularly GtACR1 and GtACR2, are increasingly used due to their large photocurrents and high sensitivity to light (*Forli et al., 2018*; *Mardinly et al., 2018*; *Mauss et al., 2017*; *Mohamed et al., 2017*; *Mohammad et al., 2017*). To use these tools to their full potentials, it is necessary that we understand their function and, importantly, their limitations. Our experiments reveal that wide-field activation of these channels in cortical neurons suppresses action potentials at the soma but also triggers neurotransmitter release at the presynaptic terminals, thereby voiding inhibition of neuronal activity. As demonstrated, the excitatory action of chloride channels in the axon and presynaptic terminals is due to the high intracellular chloride concentrations that create a depolarizing chloride electrochemical gradient. The depolarizing effect of presynaptic chloride channels, GABA$_A$ or glycine receptors, has been documented at a few synapses in the hippocampus, cerebellum, and brainstem through application of agonists or antagonists (*Jang et al., 2006*; *Pugh and Jahr, 2011*; *Ruiz et al., 2010*; *Stell et al., 2007*; *Turecek and Trussell, 2001*; *Zorrilla de San Martin et al., 2017*). Our results in cortical excitatory and inhibitory neurons expand previous findings and indicate that a presynaptic depolarizing chloride electrochemical gradient is likely a general property across brain regions and neuronal types. More importantly, our experiments directly demonstrate that activation of presynaptic chloride channels is in fact excitatory, as it can elicit action potentials with short latency, which was difficult to explicitly demonstrate by agonist application (*Pugh and Jahr, 2011*). Thus, light-gated chloride channels can be targeted to different brain regions, cell types, and subcellular compartments to study chloride electrochemical gradients with unprecedented temporal precision.

Although the excitatory effect of light-gated chloride channels is undesired for neuronal silencing, a potential utilization of their dual actions at the presynaptic terminals and soma is to activate specific projections of neurons while minimizing the effect of antidromic spikes. For example, long-range projection neurons often target multiple brain areas, and sometimes it is desired to selectively excite the axonal terminals projecting to one particular area. If ChR2 is used, local activation of

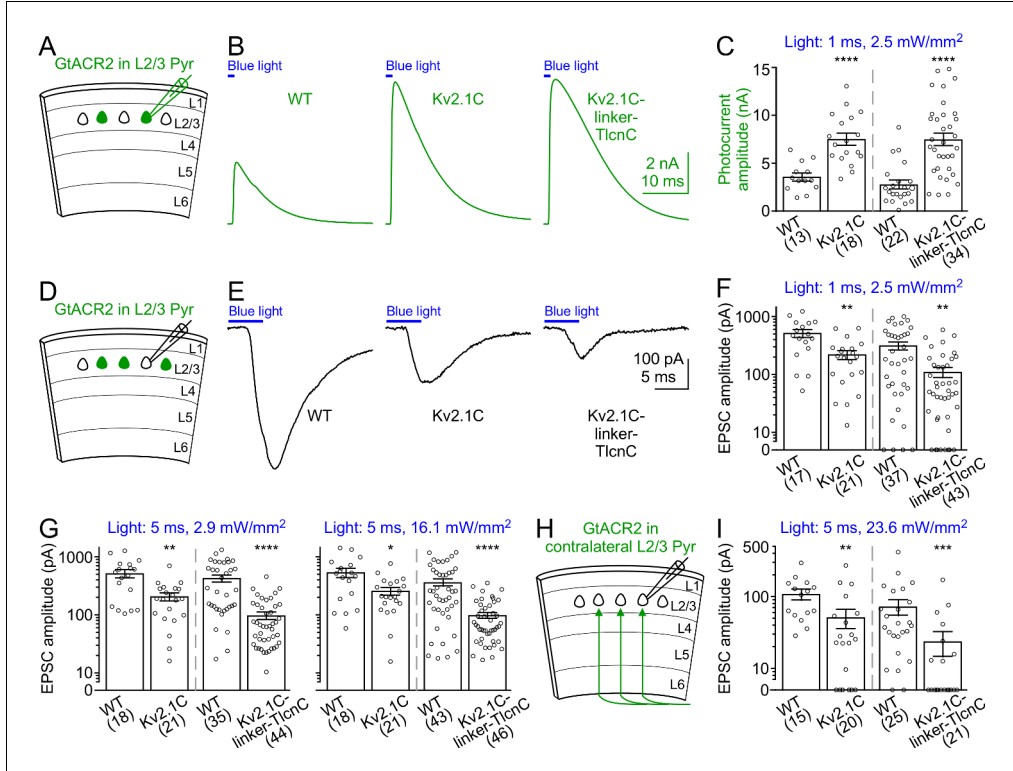

**Figure 5.** Activation of somatodendritically targeted GtACR2 variants generates larger photocurrents but causes less neurotransmitter release than wild type GtACR2. (**A**) Schematic of slice experiments in (**B,C**). GtACR2-EYFP, GtACR2-EYFP-Kv2.1C, or GtACR2-EYFP-Kv2.1C-linker-TlcnC in a subset of layer 2/3 pyramidal neurons. (**B**) Somatic photocurrents recorded at the membrane potential of +10 mV from GtACR2-EYFP-, GtACR2-EYFP-Kv2.1C-, or GtACR2-EYFP-Kv2.1C-linker-TlcnC-expressing neurons in response to a blue light pulse of 1 ms and 2.5 mW/mm$^2$. (**C**) Summary graph showing the photocurrent peak amplitudes in response to a blue light pulse of 1 ms and 2.5 mW/mm$^2$. (**D,E**) As in (**A,B**), but for recording EPSCs in GtACR2$^-$ layer 2/3 pyramidal neurons in response to a blue light pulse of 5 ms and 2.9 mW/mm$^2$. (**F,G**) Summary graphs showing the EPSC amplitudes in response to a blue light pulse of 1 ms and 2.5 mW/mm$^2$ (**F**), 5 ms and 2.9 mW/mm$^2$ (**G**, left panel), or 5 ms and 16.1 mW/mm$^2$ (**G**, right panel). Note that at the stimulation strength of 1 ms and 2.5 mW/mm$^2$ (**F**), EPSCs were not observed in a subset of neurons, particularly for GtACR2-EYFP-Kv2.1C-linker-TlcnC. (**H**) Schematic of slice experiments in (**I**). GtACR2-EYFP, GtACR2-EYFP-Kv2.1C, or GtACR2-EYFP-Kv2.1C-linker-TlcnC in a subset of layer 2/3 pyramidal neurons in the contralateral hemisphere. (**I**) Summary graphs showing the EPSC amplitudes of GtACR2$^-$ layer 2/3 pyramidal neurons in response to stimulating the callosal axons with a blue light pulse of 5 ms and 23.6 mW/mm$^2$. The numbers of recorded neurons were indicated in the panels. The columns and error bars are mean ± s.e.m. *$p<0.05$, **$p<0.01$, ***$p<0.001$, ****$p<0.0001$.

DOI: https://doi.org/10.7554/eLife.38506.015

The following figure supplement is available for figure 5:

**Figure supplement 1.** GtACR2-induced photocurrents and EPSCs at different strengths of blue light stimulation.
DOI: https://doi.org/10.7554/eLife.38506.016

---

ChR2-expressing axonal terminals may generate antidromic spikes, which will affect other projections. However, if GtACR2 is used, one can simultaneously activate GtACR2 in the soma and axonal terminals. Axonal depolarization will result in neurotransmitter release, but the antidromic spikes will be reduced or suppressed by the hyperpolarization originating from the somatodendritic domain, which reduces the likelihood of activating other projections.

To create a better inhibitory optogenetic tool, we tested a number of somatodendritic targeting motifs to confine GtACR2 in the soma and dendrites. We generated a hybrid motif, Kv2.1C-linker-TlcnC, that was more effective than the widely used Kv2.1C (*Baker et al., 2016*; *Mardinly et al., 2018*; *Wu et al., 2013*). The Kv2.1C motif was recently used in a bioRxiv preprint (*Mahn et al.,*

2017) to target GtACR2 to the somatodendritic domain. This GtACR2 variant achieved a greater reduction of neurotransmitter release from the contralateral callosal projections of medial prefrontal cortical neurons than what we observed with our GtACR2-EYFP-Kv2.1C and GtACR2-EYFP-Kv2.1C-linker-TlcnC in visual cortical neurons. This quantitative difference is likely due to different expression levels, light stimulation strengths, or neuronal types (see discussion below), which reiterates the importance of precisely characterizing the effect of light-gated chloride channels, specific to each experimental design, when using them to manipulate neuronal activity.

While GtACR2-EYFP-Kv2.1C-linker-TlcnC can still traffic to the axon to cause neurotransmitter release, it is thus far the most improved light-gated chloride channel for optogenetic inhibition. Since it can generate much larger photocurrents in the somatodendritic domain than what is necessary to suppress action potentials, one approach to use this tool is to reduce the overall GtACR2 expression level to further decrease its presence in the axon while still generating sufficient inhibitory photocurrents at the soma. Supporting this idea, *post hoc* analysis of our experiments indicates that, for a given experiment, it is possible to identify a light stimulation strength that is low enough not to cause neurotransmitter release but still produce large inhibitory photocurrents in GtACR2$^+$ neurons (*Figure 5—figure supplement 1*). However, this stimulation strength must be empirically determined for every experiment. Furthermore, since during continuous activation of GtACR2, large and synchronous neurotransmitter release occurs at the early phase of the light stimulation (*Figure 1—figure supplement 3*), one may also take advantage of the late phase when neurotransmitter release is reduced but photocurrent is still robust for optogenetic inhibition experiments. Another approach is to selectively photostimulate GtACR2 at the soma by two-photon excitation (*Forli et al., 2018*; *Mardinly et al., 2018*). However, it is difficult to apply this photoactivation approach to freely moving animals or deep brain areas. Therefore, it is imperative that we further engineer light-gated chloride channels to eliminate their excitatory action in the axonal terminals. Future strategies include generating more effective somatodendritic targeting motifs, creating outwardly rectifying channels, and the combination of both strategies. The GtACR2-EYFP-Kv2.1C-linker-TlcnC reported here enhances the available toolkit for optogenetic inhibition and will serve as the foundation for future improvement.

## Materials and methods

**Key resources table**

| Reagent type (species) or resource | Designation | Source or reference | Identifiers | Additional information |
|---|---|---|---|---|
| Strain, strain background (*M. musculus*) | *Pvalb-2A-Flpo* | Jackson Laboratory | RRID:IMSR_JAX:022730 | |
| Strain, strain background (*M. musculus*) | *Pvalb-2A-Cre* | Jackson Laboratory | RRID:IMSR_JAX:012358 | |
| Recombinant DNA reagent (plasmid) | GtACR2 | *Govorunova et al., 2015* | Addgene:67877 | pLenti-UbiC-GtACR2-EYFP; obtained from Dr. John Spudich |
| Recombinant DNA reagent (plasmid) | GtACR1 | *Govorunova et al., 2015* | Addgene:67795 | pLenti-UbiC-GtACR1-EYFP; obtained from Dr. John Spudich |
| Recombinant DNA reagent (plasmid) | iC++ | *Berndt et al., 2016* | | pAAV-CaMKIIα-iC++-TS-EYFP or pAAV-EF1α-DIO-iC++-TS-EYFP; obtained from Dr. Karl Deisseroth |

*Continued on next page*

*Continued*

| Reagent type (species) or resource | Designation | Source or reference | Identifiers | Additional information |
|---|---|---|---|---|
| Recombinant DNA reagent (plasmid) | Flpo | *Xue et al. (2014)* | Addgene:60662 | pCAG-Flpo |
| Recombinant DNA reagent (plasmid) | ChR2 | this paper | Addgene:114367 | pCAG-hChR2(H134R)-EYFP |
| Recombinant DNA reagent (plasmid) | Flpo-dependent GtACR2 | this paper | Addgene:114369 | pAAV-EF1α-FRT-FLEX-GtACR2-EYFP |
| Recombinant DNA reagent (plasmid) | Flpo-dependent GtACR1 | this paper | Addgene:114370 | pAAV-EF1α-FRT-FLEX-GtACR1-EYFP |
| Recombinant DNA reagent (plasmid) | GtACR2-EYFP-Kv2.1C | this paper | Addgene:114375 | pAAV-EF1α-FRT-FLEX-GtACR2-EYFP-Kv2.1C |
| Recombinant DNA reagent (plasmid) | GtACR2-EYFP-Kv2.1C-linker-TlcnC | this paper | Addgene:114377 | pAAV-EF1α-FRT-FLEX-GtACR2-EYFP-Kv2.1C-linker-TlcnC |
| Transfected construct (adeno-associated virus) | Flpo | *Xue et al. (2014)*; Penn Vector Core | Addgene:60663 | AAV9-hSyn-Flpo |
| Transfected construct (adeno-associated virus) | Flpo-dependent GtACR2; GtACR2 | This paper; Baylor College of Medicine Gene Vector Core | Addgene:114369 | AAV9-EF1α-FRT-FLEX-GtACR2-EYFP |
| Transfected construct (adeno-associated virus) | ReaChR | This paper; Baylor College of Medicine Gene Vector Core | Addgene:114368 | AAV9-EF1α-DIO-ReaChR-P2A-dTomato |
| Transfected construct (adeno-associated virus) | iC++ | This paper; Baylor College of Medicine Gene Vector Core | | AAV9-EF1α-DIO-iC++-TS-EYFP; plasmid obtained from Dr. Karl Deisseroth |
| Transfected construct (adeno-associated virus) | iChloC | This paper; Baylor College of Medicine Gene Vector Core | | AAV9-EF1α-DIO-iChloC-T2A-mCherry; plasmid obtained from Drs. Matthew Caudill and Massimo Scanziani |
| Chemical compound, drug | Bumetanide (Ro 10–6338) | Santa Cruz Biotechnology | Santa Cruz:sc-200727 | 50 or 100 µM |

## Mice

All procedures to maintain and use mice were approved by the Institutional Animal Care and Use Committee at Baylor College of Medicine. Mice were maintained on a 14 hr:10 hr light:dark cycle with regular mouse chow and water *ad libitum*. Experiments were performed during the light cycle. ICR (CD-1) female mice were purchased from Baylor College of Medicine Center for Comparative Medicine or Charles River Laboratories. C57BL6/J, *Pvalb-2A-Cre*, and *Pvalb-2A-Flpo* mice were obtained from Jackson Laboratory (stock numbers 000664, 012358, and 022730, respectively). Both male and female mice were used in the experiments. The mice were used at the age of 3–9 weeks for experiments, except for the conditional expression of GtACR2 in adults, where mice were used at the age of 10–12 weeks.

## DNA constructs

Plasmids pLenti-UbiC-GtACR2-EYFP (Addgene #67877) and pLenti-UbiC-GtACR1-EYFP (Addgene #67795) were obtained from Dr. John Spudich, pAAV-CaMKIIα-iC++-TS-EYFP and pAAV-EF1α-DIO-

iC++-TS-EYFP from Dr. Karl Deisseroth, pAAV-EF1α-DIO-iChloC-T2A-mCherry from Drs. Matthew Caudill and Massimo Scanziani, pCAG-tdTomato from Anirvan Ghosh, and pCAG-Cre from Addgene (#13775). Plasmid pCAG-Flpo (Addgene #60662) was previously described (*Xue et al., 2014*). All other plasmids were generated and deposited at Addgene as below. pCAG-hChR2(H134R)-EYFP (Addgene #114367) was created by replacing the EGFP in pCAG-EGFP (Addgene #11150) with the hChR2(H134R)-EYFP from pAAV-EF1α-DIO-hChR2(H134R)-EYFP (Addgene #20298). pAAV-EF1α-DIO-ReaChR-P2A-dTomato (Addgene #114368) was created by replacing the oChIEF(E163A T199C) in pAAV-EF1α-DIO-oChIEF(E163A T199C)-P2A-dTomato (Addgene #51094) with the ReaChR from pAAV-hSyn-FLEX-ReaChR-Citrine (Addgene #50955). pAAV-EF1α-FRT-FLEX-GtACR2-EYFP (Addgene #114369) and pAAV-EF1α-FRT-FLEX-GtACR1-EYFP (Addgene #114370) were created by replacing the mNaChBac-T2A-tdTomato in pAAV-EF1α-FRT-FLEX-mNaChBac-T2A-tdTomato (Addgene #60658) with the GtACR2-EYFP and GtACR1-EYFP from pLenti-UbiC-GtACR2-EYFP and pLenti-UbiC-GtACR1-EYFP, respectively. Motifs MBD and TlcnC were generated by PCR primers. Motifs Nlgn1C and Kv4.2LL were generated by PCR from pAAV-CAG-post-mGRASP-2A-dTomato (Addgene #34912) and a Kv4.2-expressing plasmid, respectively. Motif Kv2.1C was obtained from pAAV-EF1α-DIO-hChR2(H134R)-EYFP-Kv2.1C (*Wu et al., 2013*). Motifs Kv2.1C-TlcnC and Kv2.1C-linker-TlcnC were generated by PCR from Kv2.1C and TlcnC. All motifs were then added to the C-terminus of the GtACR2-EYFP to create pAAV-EF1α-FRT-FLEX-GtACR2-EYFP-MBD (Addgene #114371), pAAV-EF1α-FRT-FLEX-GtACR2-EYFP-Nlgn1C (Addgene #114372), pAAV-EF1α-FRT-FLEX-GtACR2-EYFP-Kv4.2LL (Addgene #114373), pAAV-EF1α-FRT-FLEX-GtACR2-EYFP-TlcnC (Addgene #114374), pAAV-EF1α-FRT-FLEX-GtACR2-EYFP-Kv2.1C (Addgene #114375), pAAV-EF1α-FRT-FLEX-GtACR2-EYFP-Kv2.1C-TlcnC (Addgene #114376), and pAAV-EF1α-FRT-FLEX-GtACR2-EYFP-Kv2.1C-linker-TlcnC (Addgene #114377). pAAV-EF1α-FRT-FLEX-GtACR2-KA2N-EGFP (Addgene #114378) was created by replacing the EYFP in pAAV-EF1α-FRT-FLEX-GtACR2-EYFP with the KA2N-EGFP from pAAV-hSyn-soCoChR-EGFP (Addgene #107708).

## *In utero* electroporation

Female ICR mice were crossed with male C57BL6/J, *Pvalb-2A-Cre*, or *Pvalb-2A-Flpo* mice to obtain timed pregnancies. *In utero* electroporation was performed as previously described (*Xue et al., 2014*) with a square-wave pulse generator (Gemini X2, BTX Harvard Bioscience). To express GtACR2, GtACR1, iC++, or ChR2 in layer 2/3 pyramidal neurons, pLenti-UbiC-GtACR2-EYFP, pLenti-UbiC-GtACR1-EYFP, pAAV-CaMKIIα-iC++-TS-EYFP, or pCAG-hChR2(H134R)-EYFP (all 2 µg/µl) was used, respectively. In a few experiments, pAAV-EF1α-FRT-FLEX-GtACR2-EYFP (2 µg/µl) with pCAG-Flpo (0.2 µg/µl), pAAV-EF1α-FRT-FLEX-GtACR1-EYFP (2 µg/µl) with pCAG-Flpo (0.2 µg/µl), or pAAV-EF1α-DIO-iC++-TS-EYFP (2 µg/µl) with pCAG-Cre (0.2 µg/µl) was used to express GtACR2, GtACR1, or iC++, respectively. To express somatodendritically targeted GtACR2 variants in layer 2/3 pyramidal neurons and compare them with wild type GtACR2, the pAAV-EF1α-FRT-FLEX constructs described above were used (all 2 µg/µl) with pCAG-Flpo (0.2 µg/µl). pCAG-tdTomato (0.1 µg/µl) was included in all experiments. The plasmid concentrations stated above were final concentrations in the plasmid mix. Transfected pups were identified by the transcranial fluorescence of tdTomato with a MZ10F stereomicroscope (Leica) 1–2 days after birth.

## AAV production and injection

All recombinant AAV serotype 9 vectors were produced by the Gene Vector Core at Baylor College of Medicine except AAV9-hSyn-Flpo (Addgene #60663), which was produced by the Penn Vector Core (*Xue et al., 2014*). To express GtACR2, ReaChR, iC++, or iChloC in Pv neurons, 200–250 nl of the following recombinant AAV serotype 9 vectors at their respective titer were injected into the visual cortex of $Pvalb^{Flpo/+}$ (for GtACR2) or $Pvalb^{Cre/+}$ (for ReaChR, iC++, or iChloC) mice at postnatal day 1 as previously described (*Xue et al., 2014*): AAV9-EF1α-FRT-FLEX-GtACR2-EYFP ($3.8 \times 10^{13}$ genome copies/ml), AAV9-EF1α-DIO-ReaChR-P2A-dTomato ($7.0 \times 10^{13}$ genome copies/ml), AAV9-EF1α-DIO-iC++-TS-EYFP ($3.71 \times 10^{13}$ genome copies/ml), and AAV9-EF1α-DIO-iChloC-2A-mCherry ($3.7 \times 10^{14}$ genome copies/ml). To conditionally express GtACR2 in juvenile and adult neurons, mice previously electroporated with plasmid pAAV-EF1α-FRT-FLEX-GtACR2-EYFP into layer 2/3 pyramidal neurons were injected with 200 nl of AAV9-hSyn-Flpo ($1.2 \times 10^{12}$ genome

copies/ml) at postnatal day 23, 60, or 64. Injection was performed as previously described (*Xue et al., 2014*) with an UltraMicroPump III and a Micro4 controller (World Precision Instruments).

## Brain slice electrophysiology

Mice were anesthetized by an intraperitoneal injection of a ketamine and xylazine mix (80 mg/kg and 16 mg/kg, respectively) and transcardially perfused with cold (0–4°C) slice cutting solution containing 80 mM NaCl, 2.5 mM KCl, 1.3 mM $NaH_2PO_4$, 26 mM $NaHCO_3$, 4 mM $MgCl_2$, 0.5 mM $CaCl_2$, 20 mM D-glucose, 75 mM sucrose and 0.5 mM sodium ascorbate (315 mosmol, pH 7.4, saturated with 95% $O_2$/5% $CO_2$). Brains were removed and sectioned in the cutting solution with a VT1200S vibratome (Leica) to obtain 300 µm coronal slices. Slices were incubated in a custom-made interface holding chamber containing slice cutting solution saturated with 95% $O_2$/5% $CO_2$ at 34°C for 30 min and then at room temperature for 20 min to 10 hr until they were transferred to the recording chamber.

Recordings were performed on submerged slices in artificial cerebrospinal fluid (ACSF) containing 119 mM NaCl, 2.5 mM KCl, 1.3 mM $NaH_2PO_4$, 26 mM $NaHCO_3$, 1.3 mM $MgCl_2$, 2.5 mM $CaCl_2$, 20 mM D-glucose and 0.5 mM sodium ascorbate (305 mosmol, pH 7.4, saturated with 95% $O_2$/5% $CO_2$, perfused at 3 ml/min) at 30–32°C. For whole-cell recordings, a $K^+$-based pipette solution containing 142 mM $K^+$-gluconate, 10 mM HEPES, 1 mM EGTA, 2.5 mM MgCl2, 4 mM ATP-Mg, 0.3 mM GTP-Na, 10 mM Na2-phosphocreatine (295 mosmol, pH 7.35) or a $Cs^+$-based pipette solution containing 121 mM $Cs^+$-methanesulfonate, 1.5 mM $MgCl_2$, 10 mM HEPES, 10 mM EGTA, 4 mM Mg-ATP, 0.3 mM Na-GTP, 10 mM $Na_2$-Phosphocreatine, and 2 mM QX314-Cl (295 mosmol, pH 7.35) was used. Membrane potentials were not corrected for liquid junction potential (experimentally measured as 12.5 mV for the $K^+$-based pipette solution and 9.5 mV for the $Cs^+$-based pipette solution).

Neurons were visualized with video-assisted infrared differential interference contrast imaging, and fluorescent neurons were identified by epifluorescence imaging under a water immersion objective (40×, 0.8 numerical aperture) on an upright SliceScope Pro 1000 microscope (Scientifica) with an infrared IR-1000 CCD camera (DAGE-MTI). Data were low-pass filtered at 4 kHz and acquired at 10 kHz with an Axon Multiclamp 700B amplifier and an Axon Digidata 1550 Data Acquisition System under the control of Clampex 10.7 (Molecular Devices). Data were analyzed offline using AxoGraph X (AxoGraph Scientific). For the photostimulation of GtACR2-, iC++-, iChloC-, or ChR2-expressing neurons, blue light was emitted from a collimated light-emitting diode (LED) of 455 nm, whereas for the photostimulation of GtACR1- or ReaChR-expressing neurons, red light was emitted from a LED of 617 nm. The LEDs were driven by a LED driver (Mightex) under the control of an Axon Digidata 1550 Data Acquisition System and Clampex 10.7. Light was delivered through the reflected light fluorescence illuminator port and the 40× objective.

Synaptic currents and photocurrents were recorded in the whole-cell voltage clamp mode with the $Cs^+$-based patch pipette solution. Only recordings with series resistance below 20 MΩ were included. EPSCs and IPSCs were recorded at the reversal potential for IPSCs (−60 mV) and EPSCs (+10 mV), respectively, unless stated otherwise. Photocurrents were recorded at +10 mV unless stated otherwise. For short light pulse stimulation, pulse duration (0.5–10 ms) and intensity (2.3–23.6 mW/mm$^2$) were adjusted for each recording to evoke small (to minimize voltage-clamp errors) but reliable monosynaptic EPSCs or IPSCs. Disynaptic IPSCs were evoked using the same light pulses that were used for evoking the corresponding monosynaptic EPSCs. Light pulses were delivered at 30 s interstimulus intervals. For long light pulse stimulation, blue light was delivered for 2 s at 60 s interstimulus intervals. To obtain the cumulative charge transfer curves, the EPSC traces were high-pass filtered at 0.3 Hz to minimize the effect of slow changes in the baselines, except for two neurons that were filtered at 0.5 Hz or 1.3 Hz. It was not necessary to high-pass filter the IPSC traces. The traces were baselined before the light onset and then integrated to calculate the cumulative charge transfers. Antidromic spikes in GtACR2$^+$ neurons were recorded with the K+-based patch pipette solution in whole-cell current clamp mode or with ACSF as the patch pipette solution in the loose-patch current clamp mode.

For pharmacology experiments, the baseline synaptic currents were recorded for at least 3 min in the absence of any drug. The drugs were then added to the ACSF at the following concentrations: TTX (1 µM), NBQX (10 µM), (*RS*)-CPP (10 µM), SR95531 (Gabazine, 10 µM), ZD7288 (20 µM), bumetanide (50 or 100 µM), TEA (1.5 mM), and 4-AP (1.5 mM). The synaptic currents were recorded for at least 3 min in the presence of drugs. For ZD7288, which did not inhibit GtACR2-induced

neurotransmitter release, the efficacy of the drug was monitored by examining the $I_h$ current of cortical layer 5 pyramidal neurons.

## Fluorescent microscopy

Fluorescent images were taken from live brain slices, except for the conditional expression of GtACR2 in adults, where images were taken from fixed brain slices. Live brain slices were prepared as described for slice electrophysiology. For the fixed brain slices, mice were anesthetized by an intraperitoneal injection of a ketamine and xylazine mix (80 mg/kg and 16 mg/kg, respectively) and transcardially perfused with phosphate buffered saline (PBS, pH 7.4) followed by 4% paraformaldehyde in PBS (pH 7.4). Brains were further fixed overnight in 4% paraformaldehyde, cryoprotected with 30% sucrose in PBS, and frozen in optimum cutting-temperature medium until sectioning. A HM 450 Sliding Microtome (Thermo Scientific) was used to section the brains to obtain 30–50 mm coronal slices. Images were acquired on an Axio Zoom.V16 Fluorescence Stereo Zoom Microscope (Zeiss) and processed using National Institutes of Health ImageJ.

To determine the EYFP (or EGFP) fluorescence ratio between layer 5 and layer 2/3, one or two 350 μm-wide rectangular regions that were perpendicular to the pia and spanned all 6 cortical layers were selected in the most transfected regions of each slice. The mean EYFP (or EGFP) fluorescence was measured for layer 5 and layer 2/3 within the selected area. The mean tdTomato fluorescence was measured similarly for layer 5 and layer 2/3. The mean background fluorescence was measured from a nearby rectangular region (140.5 μm by 90.8 μm) where no cellular EYFP (or EGFP) and tdTomato fluorescence was present. The normalized EYFP (or EGFP) fluorescence ratio between layer 5 and layer 2/3 was calculated by $\frac{Layer\ 5_{EYFP} - Background_{EYFP}}{Layer\ 2/3_{EYFP} - Background_{EYFP}} / \frac{Layer\ 5_{tdTomato} - Background_{tdTomato}}{Layer\ 2/3_{tdTomato} - Background_{tdTomato}}$.

To determine the ratio of EYFP fluorescence to tdTomato fluorescence in the callosal projections, one or two rectangular regions that contained the tdTomato-labeled axons were selected in each slice to measure the mean EYFP and tdTomato fluorescence. The mean background fluorescence was measured in a nearby cortical area spanning the same layers. The ratio of EYFP fluorescence to tdTomato fluorescence was calculated by $\frac{Fluorescence_{EYFP} - Background_{EYFP}}{Fluorescence_{tdTomato} - Background_{tdTomato}}$.

## Statistics

All reported sample numbers (*n*) represent biological replicates that are the numbers of recorded neurons for electrophysiology or the numbers of analyzed regions of interest (ROI) for fluorescent images. Statistical analyses were performed with Prism 7 (GraphPad Software). We first determined whether the data were normally distributed by performing the D'Agostino and Pearson test, Shapiro-Wilk test, and KS test. If all data within one experiment passed all three normality tests, we then performed the statistical test that assumes a Gaussian distribution. Otherwise, we performed the statistical test that assumes a non-Gaussian distribution. All statistical tests were two-tailed with an alpha of 0.05.

Wilcoxan matched-pairs signed rank test was used for *Figure 1F,K*; *Figure 2B,D,F,J*; *Figure 2—figure supplement 2B,E*; *Figure 2—figure supplement 3B* (EPSCs); *Figure 3B*; and *Figure 3—figure supplement 1C*. Paired *t* test was used for *Figure 1G,J*; *Figure 2H*; *Figure 2—figure supplement 1D*; and *Figure 2—figure supplement 3B* (photocurrents); *Figure 2—figure supplement 3D*. *t* test with Welch's correction was used for *Figure 5C* (WT vs. Kv2.1C). Mann-Whitney test was used for *Figure 5C* (WT vs. Kv2.1C-linker-TlcnC); *Figure 5F,G,I*; *Figure 5—figure supplement 1C,D*. Repeated measures one-way ANOVA with Greenhouse-Geisser correction and Tukey multiple comparisons with multiplicity adjusted *P* values was used for *Figure 3D,F*. Kruskal-Wallis test with Dunn's multiple comparisons with multiplicity adjusted *P* values was used for *Figure 4D*. Ordinary one-way ANOVA with Dunnet's multiple comparison test with multiplicity adjusted *P* values was used for *Figure 4E*. The details of all statistical tests, numbers of replicates and mice, and *P* values were reported in *Supplementary file 1*.

## Acknowledgments

We thank Elena Govorunova and John Spudich for discussions and the pLenti-UbiC-GtACR2-EYFP and pLenti-UbiC-GtACR1-EYFP plasmids, Anirvan Ghosh for the pCAG-tdTomato plasmid, Karl Deisseroth for the pAAV-CaMKIIα-iC++-TS-EYFP and pAAV-EF1α-DIO-iC++-TS-EYFP plasmids,

Matthew Caudill and Massimo Scanziani for the pAAV-EF1α-DIO-iChloC-T2A-mCherry plasmid, Zhuo-Hua Pan for the pAAV-EF1α-DIO-hChR2(H134R)-EYFP-Kv2.1C plasmid, Edward Boyden for the pAAV-hSyn-soCoChR-EGFP plasmid, Paul Pfaffinger for the Kv4.2-expressing plasmid, Shuyun Deng and Kazuhiro Oka at the Baylor College of Medicine Gene Vector Core for recombinant AAV vector production, and Hsiao-Tuan Chao, Matthew Caudill, and the members of the Xue Laboratory for comments on earlier versions of the manuscript. This work was supported in part by the Whitehall Foundation (grant #2015-05-54 to MX) and the National Institutes of Health (grant R01NS100893 to MX). JEM is part of the Baylor College of Medicine Medical Scientist Training Program and a McNair MD/PhD Student Scholar supported by the McNair Medical Institute at the Robert and Janice McNair Foundation. MX is a Caroline DeLuca Scholar.

## Additional information

### Funding

| Funder | Grant reference number | Author |
| --- | --- | --- |
| Whitehall Foundation | 2015-05-54 | Mingshan Xue |
| Robert and Janice McNair Foundation | MD/PhD Student Scholarship | Jessica E Messier |
| National Institutes of Health | R01NS100893 | Mingshan Xue |

The funders had no role in study design, data collection and interpretation, or the decision to submit the work for publication.

### Author contributions

Jessica E Messier, Conceptualization, Resources, Formal analysis, Validation, Investigation, Visualization, Methodology, Writing—original draft, Writing—review and editing; Hongmei Chen, Resources, Investigation; Zhao-Lin Cai, Validation, Investigation, Writing—review and editing; Mingshan Xue, Conceptualization, Resources, Formal analysis, Supervision, Funding acquisition, Validation, Investigation, Visualization, Methodology, Writing—original draft, Project administration, Writing—review and editing

### Author ORCIDs

Jessica E Messier http://orcid.org/0000-0002-5865-7043
Zhao-Lin Cai https://orcid.org/0000-0003-4034-2884
Mingshan Xue http://orcid.org/0000-0003-1463-8884

### Ethics

Animal experimentation: This study was performed in strict accordance with the recommendations in the Guide for the Care and Use of Laboratory Animals of the National Institutes of Health. All procedures to maintain and use mice were approved in the Animal Research Protocol AN-6544 by the Institutional Animal Care and Use Committee at Baylor College of Medicine.

### Decision letter and Author response

Decision letter https://doi.org/10.7554/eLife.38506.020
Author response https://doi.org/10.7554/eLife.38506.021

## Additional files

### Supplementary files

• Supplementary file 1. Statistical Reporting Table. The statistical tests, numbers of replicates and mice, and *P* values were reported for all figures and experiments.
DOI: https://doi.org/10.7554/eLife.38506.017

• Transparent reporting form

DOI: https://doi.org/10.7554/eLife.38506.018

**Data availability**

All data generated or analyzed during this study are included in the manuscript and supporting files.

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
