## [Decision Letter]

Thank you for submitting your article "Targeting light-gated chloride channels to neuronal somatodendritic domain reduces their excitatory effect in the axon" for consideration by *eLife*. Your article has been reviewed by two peer reviewers, and the evaluation has been overseen by a Reviewing Editor and Gary Westbrook as the Senior Editor. The reviewers have opted to remain anonymous. The reviewers have discussed the reviews with one another and the Reviewing Editor has drafted this decision to help you prepare a revised submission. We hope you will be able to submit the revised version within two months.

Summary:

This study explores the effects of the optogenetic tool GtACR2, an optically activated Cl channel, on cortical neurons. The results demonstrate that, owing to differences in the chloride gradient in different cellular compartments, activation of GtACR2 suppresses electrical activity in somatodendritic compartments but leads to depolarization and transmitter release from synaptic terminals. Modifying GtACR2 to target it preferentially to the somatodendritic membrane, substantially reduces light-induced neurotransmitter release. The data therefore show that Cl currents are depolarizing in cortical synaptic terminals and also provide a technical caution for experiments that seek to manipulate Cl currents for specific outcomes.

Essential revisions:

Both reviewers were highly enthusiastic about the study, including the caliber of the work, the scientific result, and the technical pointers provided. As you will see in the reviews, which are included in full below, it was noted that the somatodendritic targeting was only partially successful, and reviewer 2 therefore raised questions about the efficacy of the modified GtACR2. However, in the consultation period, it was acknowledged by both reviewers and editor that the work is of importance even without a perfect alternative. Reviewer 1's primary comment was that the study is cast primarily as an exploration of tools, while perhaps selling the work short on the actual discovery/demonstration of the depolarizing chloride gradient in a variety of cortical neurons. Given these two responses to the study, we suggest moderate revisions (1) to emphasize the scientific discovery (in the Discussion) and (2) to acknowledge the limitation of the modified GtACR2 and/or define conditions or protocols that maximize its efficacy. We realize that this parameter space may already have been maximally explored; if so, this can be indicated in the text without further experimentation.

The reviews are included in full below.

Reviewer #1:

This interesting work reports on the actions of Cl channel opsin, GtACR2, on somatodendritic vs axon/terminal domains in cortex. The essential result is that, although inhibitory to somatodendritic membrane, activation of the opsin triggers neurotransmitter release from terminals due to the depolarizing effect of Cl currents in axons and terminals. There are several strong points to this study. First, it is very well done. The sequence of logical and very clever molecular and physiological strategies revealed that a lot of thought as well as a lot of work went into the planning and execution of this study. As an aside, studies like this are often from large well-funded labs. It is impressive to see such a complete study performed in a small lab without federal funds. Second, it obviously serves as an important cautionary tale for the field regarding the use of this flavor of opsin for any study in neuroscience. Third, it points to the potency of presynaptic Cl gradients in not just depolarizing axons and terminals but actually exciting them. In this regard, I feel the authors missed an opportunity in their Discussion, which at present focuses on the work as a characterization of a tool. As the authors note, depolarizing Cl gradients have been described brainstem and cerebellum slices and in dissociated hippocampal neurons that retain little torn off terminals (the Jang study). The latter are a bit suspect as far as their physiological state. In this study we have a demonstration that not only are Cl channels depolarizing in a wide variety of cortical neurons, but that they can induce short latency, triggered action potentials. This ability to directly excite a terminal was not observed or tested explicitly for the brainstem and cerebellum, in part because authors were limited by the efficacy of agonists and how they were applied; at best, the previous work indicated that Cl channels enhance excitability in response to other stimuli. So we have here a demonstration of an important physiological principle, as well as a method for exploring it in many different brain regions. I felt the authors could have highlighted this as well in their Discussion. Overall, I am impressed, enjoyed the read, and have no technical concerns with the work.

*Reviewer #2:*

Neuroscientists hope to use light-gated chloride channels, such as GtACR2, to suppress the activity of neurons. Messier and colleagues describe a major limitation with GtACR2, namely that it is depolarizes boutons resulting in the release of neurotransmitter rather than the suppression of neuronal activity/transmission as intended. They show this happens in both excitatory and inhibitory neurons and is because the chloride reversal potential is relatively depolarizing in boutons as compared to the somatodendritic compartment of the cell. Finally, they fuse several different targeting sequences to the channel with the goal of localizing GtACR2 to the somatodendritic compartment of the neuron. The intention of this manipulation is to bias the effects of the channel to hyperpolarizing neurons, rather than depolarizing boutons and triggering release. They are moderately successful in achieving this goal.

This study is carefully executed and the experiments/interpretation are unambiguous. I am fully convinced of the limitations of GtACR2 and the underlying mechanism the authors propose. This information is useful to the community and should be published.

I am less convinced, however, that they have solved the problem. Fusing Kv2.1C-linker-TlcnC targeting motifs to the channel reduces neurotransmission associated with light activation of the channel by ~3-5 fold, but does not eliminate it. In light of this, I'm not sure that this tool is ultimately any more useful than the wild type channel.

The authors suggest reducing channel expression levels or 2-photon activation of the channel may further reduce excitation associated with channel gating (or more precisely bias the cell to hyperpolarization of the somatodendritic compartment rather than depolarization of the axon). This should be tested. Additionally, maybe there is a light intensity that could achieve this same goal? (Editor's comment: This is the point that reviewers/editor agreed was optional, since it seems likely that the result has already been optimized. However, if experimentation to address this point is feasible, please do so. If not, please discuss.)

---

## [Author Response]

Reviewer #1:[…] Third, it points to the potency of presynaptic Cl gradients in not just depolarizing axons and terminals but actually exciting them. In this regard, I feel the authors missed an opportunity in their Discussion, which at present focuses on the work as a characterization of a tool. As the authors note, depolarizing Cl gradients have been described brainstem and cerebellum slices and in dissociated hippocampal neurons that retain little torn off terminals (the Jang study). The latter are a bit suspect as far as their physiological state. In this study we have a demonstration that not only are Cl channels depolarizing in a wide variety of cortical neurons, but that they can induce short latency, triggered action potentials. This ability to directly excite a terminal was not observed or tested explicitly for the brainstem and cerebellum, in part because authors were limited by the efficacy of agonists and how they were applied; at best, the previous work indicated that Cl channels enhance excitability in response to other stimuli. So we have here a demonstration of an important physiological principle, as well as a method for exploring it in many different brain regions. I felt the authors could have highlighted this as well in their Discussion. Overall, I am impressed, enjoyed the read, and have no technical concerns with the work.

Reviewer #1 is correct that most of the previous studies on the presynaptic GABA_A_ or glycine receptors in the brainstem and cerebellum did not directly demonstrated that activation of these chloride channels can trigger action potentials, except that Pugh et al. (2011) observed antidromic spikes when GABA was applied to the axons of some cerebellar granule cells. Nevertheless, the powerful excitatory effect of presynaptic chloride channels in cortical excitatory and inhibitory neurons was not demonstrated in previous studies. Our study also indicates that the light-gated chloride channels are excellent tools for investigating the chloride gradients in different brain regions, cell types, and subcellular compartments due to its fast kinetics and large channel conductance. We revised the text to highlight these Discussion points.

Reviewer #2:[…] I am less convinced, however, that they have solved the problem. Fusing Kv2.1C-linker-TlcnC targeting motifs to the channel reduces neurotransmission associated with light activation of the channel by ~3-5 fold, but does not eliminate it. In light of this, I'm not sure that this tool is ultimately any more useful than the wild type channel.The authors suggest reducing channel expression levels or 2-photon activation of the channel may further reduce excitation associated with channel gating (or more precisely bias the cell to hyperpolarization of the somatodendritic compartment rather than depolarization of the axon). This should be tested. Additionally, maybe there is a light intensity that could achieve this same goal? (Reviewing Editor's comment: This is the point that reviewers/editor agreed was optional, since it seems likely that the result has already been optimized. However, if experimentation to address this point is feasible, please do so. If not, please discuss.)

Reviewer #2 is correct that the somatodendritically targeted GtACR2 reduces, but does not eliminate, the excitatory effect in the axons, as we stated in the manuscript that “GtACR2-EYFP-Kv2.1C-linker-TlcnC can still traffic to the axon to cause neurotransmitter release”. After we discovered the excitatory effect of GtACR2 in the axon and presynaptic terminals in 2015, we have explored many different ways to minimize this undesired effect including various light stimulation protocols. In the revised manuscript, we added a new figure (Figure 5—figure supplement 1) to show the GtACR2-mediated photocurrents and EPSCs at different light stimulation strengths. Our results indicate that it is possible to identify a light stimulation strength that is low enough not to cause neurotransmitter release but still produce large inhibitory photocurrents in GtACR2^+^ neurons. However, this stimulation strength must be empirically determined for every experiment. Furthermore, we added another new figure (Figure 1—figure supplement 3) to show the time course of neurotransmitter release during a long pulse of light. Since large and synchronous neurotransmitter release occurs at the early phase of the light stimulation, one may take advantage of the late phase when neurotransmitter release is reduced but photocurrent is still robust for optogenetic inhibition experiments. These additional results were also discussed.